# Integrating Biochemical and Computational Approaches Reveal Structural Insights in Trastuzumab scFv-Fc Antibody Engineering

**DOI:** 10.3390/biom15050606

**Published:** 2025-04-22

**Authors:** Olga Bednova, Jessica Pougoue Ketchemen, Hazem Mslati, Mark Barok, Heikki Joensuu, Natalie Zeytuni, Francesco Gentile, Leon Sanche, Humphrey Fonge, Jeffrey Victor Leyton

**Affiliations:** 1Département de Médecine Nucléaire et Radiobiologie, Faculté de Médecine et des Sciences de la Santé, Université de Sherbrooke, Sherbrooke, QC J1H 5N4, Canada; leon.sanche@usherbrooke.ca; 2Faculté de Pharmacie, Université Laval, Québec, QC G1V 0A6, Canada; jessica.pougoue-ketchemen@crchudequebec.ulaval.ca (J.P.K.); humphrey.fonge@crchudequebec.ulaval.ca (H.F.); 3Axe Oncologie, Centre de Recherche du CHU de Québec-Université Laval, Québec, QC G1J 5B3, Canada; 4Department of Cellular and Molecular Medicine, Faculty of Medicine, University of Ottawa, Ottawa, ON K1H 8M5, Canada; hmsla015@uottawa.ca; 5 Helsinki University Hospital and University of Helsinki, 00290 Helsinki, Finland; barok.mark@gmail.com (M.B.); heikki.joensuu@hus.fi (H.J.); 6Laboratory of Molecular Oncology, Biomedicum, University of Helsinki, 00290 Helsinki, Finland; 7Department of Oncology, Helsinki University Hospital and University of Helsinki, 00029 Helsinki, Finland; 8Department of Anatomy and Cell Biology, McGill University, Montreal, QC H3A 0C7, Canada; 9Centre de Recherche en Biologie Structurale (CRBS), Montreal, QC H3G 0B1, Canada; 10Department of Chemistry and Biomedical Sciences, University of Ottawa, Ottawa, ON K1H 8M5, Canada; fgentile@uottawa.ca; 11School of Pharmaceutical Sciences, Faculty of Medicine, University of Ottawa, Ottawa, ON K1H 8M5, Canada

**Keywords:** trastuzumab, antibody engineering, HER2, scFv-Fc, molecular dynamic simulation

## Abstract

Antibody-based agents have become a preferred treatment for various diseases, including cancer, due to significant advances in antibody engineering. The use of single-chain Fv-Fcs (scFv-Fcs) has been a promising engineering approach for therapeutic design. The concept is that the Fc provides increased stability and target binding and ultimately improves performance. However, the structural and dynamic relationship between the variable and Fc domains, which are fused in close proximity, and the impact on stability and target binding are not well understood. This study evaluated trastuzumab-derived scFv-Fc antibodies, focusing on the impact of their design on important biopharmaceutical parameters. Computational modelling and molecular dynamics, alongside experimental studies, were used to ascertain their dynamics, expression and purification, stabilities, and binding potencies. The results showed that the scFv subunits exhibited stochastic interplays that lead to diverse shapes and were associated with functional performance. This new understanding of scFv-Fc antibodies and their structural and functional nuances provides important details to further guide the design of more effective and less toxic therapeutics.

## 1. Introduction

Monoclonal antibodies (mAbs), which represent powerful tools to target antigens, gained prominence over the past 30 years as a preferred treatment option for diverse diseases, including cancer. The Antibody Society reports that approximately 135 antibody-based pharmaceuticals have been approved by the United States Food and Drug Administration [1]. This success has been due to the significant advancements in antibody engineering, such as humanization, phage-display technologies, and genetic fusions. Such engineering often induces profound changes in a protein’s structure and, therefore, a careful consideration of the three-dimensional (3D) spatial orientations of the entire protein or its individual domains is crucial. This focus on 3D structure is important in optimizing stability and performance, and for propelling the continued demand for engineered antibodies due to their enhanced therapeutic potential.

The single-chain variable fragment (scFv) consists of only the variable light (V_L_) and variable heavy (V_H_) domains, which are genetically fused by a polypeptide linker. The original pharmaceutical rationale for developing the scFv was to maintain the potent targeting and inhibitory activity of a mAb while producing high yields at a much lower cost compared to mAbs [2]. Despite the advantages of scFvs, they have a few drawbacks that limit their therapeutic potential. First, scFvs tend to aggregate both in vivo and in vitro and are sensitive to temperature [3,4,5]. As a single unit, the scFv has previously exhibited reduced stability as the V_L_ and V_H_ domains separate and rebind frequently and aggregate under small stresses [6]. Second, because scFvs are approximately 25 kDa, they have an extremely short serum half-life of <1 day compared to days and weeks for immunoglobulin G (IgG) 1 isotype-derived mAbs [7]. Third, scFvs have reduced affinity compared to the parental mAb counterparts [8]. Taken together, these drawbacks are the principal causes for the limited number of scFv-approved therapeutics, particularly those that must be administered intravenously [2].

One difficulty for scFv antigen binding stems from the intrinsic structural orientation of the antigen recognition domains of a mAb. Although the sequence diversity in antibodies is crucial for antigen specificity, the overall folds of the V_L_ and V_H_ domains are well conserved. The differences in structure within each variable domain are largely influenced by the complementarity-determining regions (CDRs), which are the six hypervariable loops presented on the ‘tips’ of the structurally conserved frameworks that are essential for antigen binding. Additionally, since an antibody paratope is located at the interface of the V_L_ and V_H_ domains, the relative spatial orientations of the CDRs can be influenced by the association of the V_L_/V_H_ heterodimer and can significantly impact the antigen binding properties [9].

Most currently approved antibodies are those of the IgG_1_ isotype [10]. The structure of an IgG_1_ comprises two antigen-binding Fab arms linked to a single fragment crystallizable (Fc) domain via a hinge region. This structural design enables antibodies to bind target antigens bivalently while simultaneously linking with humoral and cellular components of the immune system. The cellular immune responses occur mostly due to the interactions between the Fc and Fc-specific receptors on the surface of immune cells, activating signalling and leading to effector functions such as antibody-dependent cellular cytotoxicity and complement-dependent cytotoxicity [10]. Notably, the Fc is well known to significantly contribute to the overall stability of an IgG [11]. The structure of the Fc reveals how the two constant domains, C_H_2 and C_H_3, of each heavy chain interact with one another. The C_H_3 domains pack tightly with each other, while the C_H_2 domains have no observable protein–protein contact with each other.

The success of mAb-based therapies is also significantly based on their long circulating half-life, which is dependent on the Fc. The Fc controls the serum and tissue half-life through interactions with the neonatal Fc receptor (FcRn) [12,13], and it was originally named based on historical experiments that determined that the FcRn has a major role in mediating the transfer of humoral immunity from mother to neonates [14]. During IgG circulation in the bloodstream, they are randomly internalized in cells by pinocytosis and contained in the intracellular environment in endosomes, where FcRn is primarily expressed [10]. IgG_1_ antibodies contain conserved histidine residues at the C_H_2-C_H_3 domain interface that are neutral at a physiologic pH of 7.4. But when exposed to the acidic endosomal environment (pH < 6.5), the histidine residues become protonated and enable the Fc to bind to FcRn [15]. The IgG-FcRn interaction triggers endosome transport to the cell surface and upon fusion with the plasma membrane and exposure to the extracellular neutral pH environment, the antibody is released. Our current understanding is that this antibody capture and release mechanism by FcRn protects IgGs from lysosomal catabolism and results in their long circulating half-life of 7–21 days depending on the IgG subtype [16].

For many researchers, the challenge in antibody-based therapeutics lies in antibody conjugate design for delivering cytotoxic payloads. Although unmodified mAbs show some therapeutic potency, their effects tend to be various and ultimately not curative [17]. For mAbs conjugated to cytotoxic payloads, the long circulating half-life of antibodies is advantageous due to the direct association with target tissue uptake and therapeutic efficacy. There is also a direct causal relationship with increased systemic toxicity. Despite the remarkable advancement of these types of agents such as antibody–drug conjugates (ADCs) with increasing approvals for diverse cancers, there are still serious issues concerning toxicity, which are directly related to prolonged IgG circulation half-life [18,19].

The Wu group pioneered the development of several engineered antibody formats based on the scFv unit fused to various components of the hinge and Fc of IgG_1_. Specifically, they demonstrated that formats above the threshold for first-pass renal clearance show intermediate clearance rates and reach moderate tumour uptake levels for therapeutic applications with radioactive- or photothermal-emitting payloads [20,21,22]. One format known as the scFv-Fc (110–120 kDa) has been shown to have good properties such as high tumour uptake, comparable to mAbs, but it has slightly faster blood clearance due to its smaller size [23]. Additionally, Kenanova et al. demonstrated that mutating the conserved histidine residues H310 and H435 at the FcRn binding site of the Fc to alanine (H310A) and glutamine (H435Q) resulted in accelerated blood clearance [23]. Importantly, the tumour uptake of the mutant scFv-Fc maintained high tumour uptake relative to the parental intact IgG. Additionally, by comparing scFv-Fc with either the wild-type (WT) Fc, H435Q alone (herein termed single mutant (SM)), or H435Q/H310A (herein termed double mutant (DM)), the tumour uptake–blood clearance relationship could be tailored. Ultimately, this provides an opportunity to obtain the rare and desired combination of high tumour-to-background ratios and high absolute tumour uptake required for antibody conjugate therapeutics but at significantly earlier times after administration, thereby potentially minimizing unwanted toxicities and boosting efficacy.

Therefore, the Fc has been the focus of significant engineering to modulate effector function activities. It is well accepted that the mosaic nature of engineered antibody formats exhibits impaired stability when compared to the intact parental antibody, which is owing but not limited to V_H_-V_L_ pairing and potential clashes between different IgG domains that would typically not interact in the full-sized format [11]. These are serious issues as they can impair high-affinity antigen binding and thermostability, which are essential for success in the clinic. However, the scFv and Fc relationship in the context of the constructed scFv-Fc format is currently unknown.

This work evaluated the impact of engineering on the shape and stability of trastuzumab scFv-Fc and its associations with expression, purification, and binding against human epidermal growth factor 2 (HER2). Specifically, the WT, SM, and DM scFv-Fc antibodies were generated, and their stability was measured using biochemical and computational methods and linked to their expression yields and final purities. Their performance was measured by flow cytometric analyses for binding HER2-positive JIMT-1 cells. Our findings revealed that these scFv-Fc antibodies have stochastic and markedly different spatial structural orientations that impact their overall stabilities and binding activities, and that did not involve the amino acid mutations. Importantly, the scFv subunits drove instability through direct interchain interactions, causing large-scale scFv-Fc structural deformations. If abundant expression and stable fractions were obtained, the DM scFv-Fc antibody retained the least structural reorganization and was biochemically associated with excellent dimerization and nanomolar affinity for HER2 when compared to trastuzumab.

## 2. Materials and Methods

### 2.1. Cell Lines

The human breast cancer cell line JIMT-1 (CVCL-2077) with HER2 overexpression was cultured in monolayers by serial passages in high glucose Dulbecco’s Modified Eagle’s medium (DMEM) (Hyclone Laboratories, Logan, UT, USA). The media was supplemented with 10% fetal bovine serum (FBS) (Biochrom, Sigma-Aldrich, St Louis, MO, USA) and 1% penicillin–streptomycin (Hyclone Laboratories). This cell line was authenticated using short tandem repeat profiling at the Centre for Applied Genomics (Hospital for Sick Kids, Toronto, ON, Canada). The human breast cancer SK-BR-3 cell line (Cedarlane, Burlington, ON, Canada) was cultured in McCoy’s 5a Medium (Wisent, St-Bruno, QC, Canada) supplemented with 10% FBS and 2 mM L-glutamine (Wisent, Saint-Jean-Baptiste, QC, Canada). Human embryonic kidney HEK293T (CRL-3216) and Chinese hamster ovary CHOK1 (CCL-61) cell lines (Cedarlane) were cultured in DMEM/F12 medium (Wisent) supplemented with 10% FBS (Wisent). ExpiCHO-S cells were cultivated in serum-free ExpiCHO Expression Medium (ThermoFisher, Waltham, MA, USA), following the manufacturer’s recommendations. All cells were free of mycoplasma prior to use.

### 2.2. scFv-Fc Genetic Engineering Design

Trastuzumab variable heavy (V_H_) and light (V_L_) chains were amplified from pVITRO1-Trastuzumab-IgG1/κ (Plasmid #61883; Addgene, Watertown, NY, USA) plasmid and linked using a (GGGGS)_3_ linker. The human IgG_1_ Fc coding sequence (GenBank: KY053479.1) was synthesized by Integrated DNA Technologies (Coralville, IA, USA) and fused downstream of the VEPKSC linker and a mutated (underlined) human IgG_1_ hinge DKTYTCPPCP (Figure 1a). The H/Y substitution in the hinge was used as this was previously shown to protect from degradation during mammalian expression [24]. The V_H_-V_L_-Fc coding wild-type Fc expression cassette was fused under the control of an interleukin-2 secretion signal sequence. A 6x-histidine sequence was fused at the C-terminus. Desired mutations in the Fc region were further introduced using overlapping polymerase chain reaction (PCR) and with the following primers: 5′-GAGGCTCTGCACAACCAGTACACGCAGAAG-3′ for H435Q and 5′-GTCCTCACCGTCCTGGCCCAGGACTGGCTGAATG-3′ for H310A mutations, respectively.

All cDNA fragments were fused by overlapping PCR using the NEBuilder HiFi DNA Assembly Cloning Kit (New England Biolabs, Ipswich, MA, USA) and cloned into pCIneo plasmid backbone (Catalog # E1841, Promega, Madison, WI, USA) for further expression in mammalian cell lines. *Escherichia coli* DH5α cells were transformed with the plasmids encoding the different antibodies. Transformed cells were plated on LB agar plates containing 100 µg/mL carbenicillin and incubated overnight at 37 °C. Recombinant plasmids containing the scFv-Fc genetic constructs were purified using the E.Z.N.A.^®^ Endo-free Plasmid DNA Mini Kit I (Omega Bio-tek, Norcross, GA, USA) and sequenced for correct gene fusion sequence identification.

### 2.3. Expression and Purification of scFv-Fc Antibodies

Adherent cellular expression was performed using CHOK1 and HEK293T cells. Prior to transfection, the cells were seeded in 100 mm plates (5 × 10^6^ cells for CHOK1 and 7.5 × 10^6^ cells for HEK293T) overnight. When cells reached ~75% confluency, the media was replaced with the same media absent of FBS. A transfection mixture was prepared according to the Lipofectamine 3000 transfection reagent instruction. Notably, Lipofectamine 3000 was mixed with Opti-Mem serum-reduced media (ThermoFisher) containing 14.75 µg of plasmid DNA. Twenty-four hours after transfection, cell media was replaced with normal culture media and expression was allowed to continue for three, five, and seven days, at which time the cells were centrifugated at 4000× *g* for 10 min at 4 °C, and then the supernatant was collected and passed through a 0.22 µm filter.

Suspension cellular expression was performed by using the ExpiCHO™ expression system (ThermoFisher) following the manufacturer’s recommendations. Twenty-four hours before transfection, ExpiCHO-S cells were diluted to a final density of 3 × 10^6^ cells/mL. The following day, the cells were counted and transferred into a fresh 125 mL Erlenmeyer flask containing 25 mL of prewarmed ExpiCHO expression medium to reach a final density of 6 × 10^6^ cells/mL. Then, 25 ug (1 µg per 1 mL of medium) of plasmid was added to 1 mL of cold OptiPRO medium. Separately, 80 µL of ExpiFectamine was diluted in 920 µL of OptiPRO medium. The two solutions were then combined and incubated at room temperature for 5 min, and then slowly transferred to the flask containing the ExpiCHO-S cells. The cells were incubated at 37 °C in a humidified atmosphere of 8% CO_2_ on an orbital shaker set to 125 revolutions/min. The next day, 150 µL of ExpiFectamine CHO Enhancer and 6 mL of ExpiCHO Feed were added to the expression flask. After eight days, supernatants were harvested as previously mentioned.

Antibodies were purified using HisPur Cobalt Superflow Agarose (ThermoFisher). Collected media was diluted 4 times using equilibration buffer (20 mM sodium phosphate, 300 mM sodium chloride, and 5 mM imidazole; pH 7.4) and incubated with cobalt beads at 4 °C on a rotator overnight. The next day, the samples were centrifuged at 700× *g* for 2 min. The supernatants were carefully removed and discarded. The resin was washed in two resin-bed volumes of wash buffer (20 mM sodium phosphate, 300 mM sodium chloride, and 10 mM imidazole; pH 7.4). Then, the samples were centrifuged for two min at 700× *g* and the supernatants were carefully removed. Next, three washing steps were performed and the antibody fragments were eluted by suspending the resin bed for 10 min in one resin-bed volume of elution buffer (20 mM sodium phosphate, 300 mM sodium chloride, and 150 mM imidazole; pH 7.4). The samples were centrifuged for 2 min at 700× *g* and the supernatants were collected and concentrated using 10 kDa cut-off Amicon Ultra-0.5 mL Centrifugal Filters (MilliporeSigma, St Louis, MO, USA).

The scFv-Fc genetic constructs were also subcloned into proprietary expression vectors by ABclonal (Woburn, MA, USA), which then transfected Expi293F cells (Thermo Fisher) using PEI MAX (Polysciences, Warrington, PA, USA). The supernatant was separated after incubation in Expi293 expression medium at 37 °C for five days, and proteins were purified with Ni-Agarose (Cube Biotech, Monheim am Rhein, Germany).

### 2.4. Purity Determination

SDS-PAGE was initially performed to evaluate antibody purities based on molecular weight (m.w.). The samples were denatured at 98 °C for three min and 10 µg of total protein was loaded onto 7.5% Mini-PROTEAN TGX Precast Protein Gel (Bio-Rad, Hercules, CA, USA). The samples were electrophoresed under reducing (dithiothreitol (DTT)) and non-reducing conditions and stained using Coomassie blue.

Antibody purities and integrities were also determined using size-exclusion high-performance liquid chromatography (SEC-HPLC) (Waters 2796 Bioseparations Module, Waters 2487 Dual λ Absorbance Detector, Xbridge^®^ BEH 200 A SEC 3.5 µm 7.8 × 150 nm column; Waters Corporation, Milford, MA, USA). The UV detector was set at 220 nm and 280 nm. The mobile phase flow rate was 0.25 mL/min for 40 min. Elution times of the scFv-Fcs were compared to the parental trastuzumab. An Agilent 2100 Bio-analyzer system (Agilent High Sensitivity Protein 250 Kit- Catalog # 5067-1575, Santa Clara, CA, USA) was utilized to further confirm purities and m.w. values, following the manufacturer’s protocol.

### 2.5. Differential Scanning Fluorimetry

The thermal stability of the scFv-Fc antibodies was evaluated by differential scanning fluorimetry (DSF) and performed using the CFX96 Real-Time PCR System (BioRad, Hercules, CA, USA). For each measurement, 1 mg/mL concentrations of each scFv-Fc were suspended in 25 µL PBS, pH 7.0, containing 2.5 µL of 50× SYPRO Orange Protein Gel stain. The tubes were heated from 10 °C to 95 °C at five °C intervals for 65 min and the resulting fluorescence data were collected. The data were then transferred to GraphPad Prism Version 9 (GraphPad, San Diego, CA, USA) for analysis by plotting the negative derivative of relative fluorescence versus temperature. The temperature of the maximal fluorescent value of the first derivative of the unfolding event is the Tm and each condition was prepared in triplicate.

### 2.6. Binding Kinetics by Flow Cytometry

JIMT-1 cells were incubated with the antibodies at increasing concentrations (12 points following threefold serial dilutions) in triplicates in 100 µL of PBS for 30 min at 4 °C. The cells were then twice washed in PBS and then suspended in a 1:100 dilution of fluorescently labelled secondary antibody goat anti-human IgG Fc (Catalog # 12-4998-82, ThermoFisher). The cells were then washed as previously described, suspended in 200 µL of PBS, and analyzed using a CytoFLEX flow cytometer (Beckman Coulter, Indianapolis, IN, USA). The flow cytometric data were analyzed using FlowJo (version 10.7.2; FlowJo LLC, Ashland, OR, USA). GraphPad was used to determine the binding dissociation constant (KD), maximum binding capacity (Bmax), and the half-maximal effective concentration (EC50) values.

### 2.7. In Silico scFv-Fc Building

The scFv-Fc antibodies were built by first loading the amino acid sequence of the scFv alone into a Molecular Operating Environment (MOE; Chemical Computing Group, Montreal, QC, Canada) and utilizing the antibody modeller feature set for ‘scFv’ that contains several trastuzumab Fab structures. A human IgG_1_ crystal structure (PDB: 1HZH [25]) was imported into MOE to represent the human IgG_1_ Fc segment as the amino acid sequence was identical to the Fc of the scFv-Fc WT. The modelled trastuzumab scFvs were then attached to the Fc via the amino acids, VEPKSC, which were manually inserted to connect with the N-terminal portion of the human IgG_1_ hinge–Fc. The H435Q and H435Q/H310A substitutions were made for the SM and DM scFv-Fc antibodies, respectively. The final in silico-generated scFv-Fc antibodies were inspected for correct amino acid sequences and protonation states at pH 7.4, and then the proteins underwent energy minimization.

### 2.8. Molecular Dynamic (MD) Simulations, RMSD, RMSF, Rg, and Structural Angle Changes

MD simulations were performed on the scFv-Fc antibodies using the Desmond MD package (version 13.7.125) [26]. The systems were minimized using the PrepWizard utility [27] and set up for explicit solvent simulations. The virtual antibodies were individually placed in an orthorhombic box containing TIP3P water and 1.15 M NaCl and the box dimensions were reduced to 76 nm × 43 nm × 86 nm, leaving 1.5 nm separation from the scFv-Fc and the box edges on all sides, and visually confirmed using Visual Molecular Dynamics [28]. The simulations were performed for 220 nanoseconds (ns) at 310 K, 1 atm. For each scFv-Fc, all frames were aligned to their initial energy-minimized structure on the ɑ-carbon atoms.

The Python MD Analysis library (version 2.9.0) [29] was used to analyze the MD trajectories for each engineered antibody using the minimized starting structure and the clustered MD configuration from the simulation. These structures were used to determine root mean square deviation (RMSD), root mean square fluctuation (RMSF), and radius of gyration (Rg) changes over time. Clustered conformations were calculated using Desmond Trajectory Clustering, setting the maximum clusters to 1 [26]. MD movies were rendered using VMD [28]. The equilibrated 220 ns trajectories were loaded and the timestep (dt) was set to 10 over the 1003 frames.

To contextualize the spatial reorganization of the scFv-Fc antibodies during MD simulations, structural angular changes were analyzed relative to their starting energy-minimized conformations. The WT, SM, and DM scFv-Fc MD-clustered structures were compared to their respective initial counterparts to quantify conformational shifts. A central pivot point and reference center points were established as outlined in Figure 1b, providing a method for interpreting structural changes. This approach was essential for three key reasons. First, the alignment of the Fc regions between the starting and MD-clustered conformations revealed a deviation of ≤2 Å, confirming that the Fc domain remained relatively stable throughout the simulation. Second, using defined center points enabled a more consistent and reliable method for quantifying angular changes, treating each scFv as a discrete structural unit. Third, given the pronounced spatial reorganization observed in the scFv and hinge regions, the N-terminal residues (P238) of both Fc chains were designated as the central pivot point to provide a standardized reference for measuring structural displacement. In addition, conformational changes in the z-axis (Å) of each scFv were measured relative to the pivot point, capturing the extent of spatial reorganization over the course of the MD simulations. This approach provided a comprehensive structural analysis, elucidating the dynamic fluctuations that may influence antigen binding and overall scFv-Fc stability.

For evaluating changes in spatial organizations of the CDRs, the scFv1 and scFv2 subunits of the MD-clustered scFv-Fc configurations were aligned onto the crystal structure of an unbound trastuzumab Fab fragment (PDB file 6BHZ [30]). The C_L_ and C_H_1 domains were deleted prior to alignment. The scFv1 and scFv2 energy-minimized structures were aligned onto the trastuzumab V_L_/V_H_ domains prior to MD simulations to ensure alignments of RMSD < 1 Å. The scFv-clustered MD conformations were then aligned to their original energy-minimized counterpart structures and the RMSD changes were determined.

## 3. Results

### 3.1. scFv-Fc V_L_ and V_H_ Orientation

The orientation of genes encoding the V_H_ and V_L_ domains can significantly affect the binding ability of engineered antibody fragments [31,32,33,34]. Typically, a V_H_-V_L_ orientation is preferred for scFv-based antibodies as the orientation of the V_H_ CDR loop 3 has a significant role in antigen binding, and it is positioned at an appropriate distance away from the flexible linker [9,35]. Additionally, fragments using a V_L_-V_H_ scFv orientation were destabilized and their binding affinity was impaired due to excessive flexibilities [36].

We observed that the WT scFv-Fc constructed in the V_H_-V_L_ orientation could be expressed and purified as a homogenous species of approximately 110 kDa. In contrast, the WT scFv-Fc constructed in the V_L_-V_H_ orientation formed large m.w. aggregates (Appendix A). Therefore, engineering continued with scFv domains only in the V_H_-V_L_ orientation.

### 3.2. scFv-Fc Expression

Different transient mammalian expression systems were explored for generating the scFv-Fc antibodies. Initial attempts at transient expression in adherent HEK293T and CHOK1 cells yielded very low protein (Table 1). In contrast, the suspension cell ExpiCHO-S system demonstrated relatively substantially higher protein expression levels, albeit modest, ranging from 6 to 19 µg/mL in a 30 mL culture volume. There were no significant differences in protein expression yields between 8- and 10-day post-transfection protocols, nor by decreasing the temperature to 32 °C. Cloning the expression cassettes into a commercial expression vector followed by transfection in 293F cells resulted in marked improvement with yields ranging from 700 to 850 µg/mL (Table 1). Individually, the DM scFv-Fc antibody exhibited a more robust expression, and higher yields were obtained across the different expression systems.

### 3.3. Purification and Biochemical Characterization

Anti-Fc affinity chromatography, using a Protein A column, resulted in severe degradation of all fragments (Appendix A). Interestingly, when the purification procedure was switched to cobalt beads, the three scFv-Fc antibodies were obtained mostly as intact dimers. SDS-PAGE gel under reducing conditions confirmed that all monomer chains migrated corresponding to an m.w. of ~55 kDa while the dimer antibody fragments migrated corresponding to m.w. values of ~120 kDa (Figure 2a). The SM scFv-Fc antibody exhibited the largest proportion of dimers at 71.9%, while the WT and DM scFv-Fc dimer populations were 64.4% and 52.5%, respectively (Figure 2a). The reference trastuzumab was recovered at 81.4% in the correct dimer population. Additionally, the fragments were stored at 4 °C for five weeks and analyzed using a bioanalyzer to determine m.w. more precisely. The purified intact dimers remained stable (Figure 2b).

The analysis of the scFv-Fc antibodies was extended to include SEC-HPLC. The elution time of trastuzumab was approximately 15.2 min and was typical for a 150 kDa mAb. In comparison, the WT, SM, and DM scFv-Fc antibodies exhibited elution times of 17.3, 17.6, and 17.7 min, respectively (Appendix A), and aligned with the expected elution times for scFv-Fc fragments with an estimated m.w. of ~123 kDa. Notably, smaller-sized protein fractions were also observed, potentially indicative of degraded components of the antibody fragments from the monomer fractions.

Therefore, these findings indicated that, although difficult, these engineered scFv-Fc agents could be expressed in robust quantities, with unique stability profiles, while keeping durable dimer populations under the present conditions needed for their characterization.

### 3.4. scFv-Fc Antibody Stability by DSF

In the stability experiments using DSF, we observed distinct melting temperatures (Tm) for the WT, SM, and DM scFv-Fc antibodies (Figure 3).

The Tm values recorded for each variant were 60.6 °C, 57.5 °C, and 59 °C, respectively, while the Tm for trastuzumab was previously reported at 80 °C, pH 7.0 [37]. The closeness in Tm values among the engineered antibodies suggested that there was no relatively significant difference in overall thermal stability among the scFv-Fc agents, but all were less stable than intact trastuzumab.

### 3.5. Binding Kinetics

Under receptor-saturating conditions, all three scFv-Fc antibodies demonstrated the ability to bind HER2 comparable to trastuzumab (Figure 4a).

When HER2-overpressing JIMT-1 cells were incubated with the scFv-Fc antibodies at increasing concentrations, the measured fluorescence showed dose-dependent binding for all agents (Figure 4b). The K_D_, Bmax, and EC_50_ values are listed in Table 2. Trastuzumab displayed the strongest affinity by having the lowest K_D_, Bmax, and EC_50_ values. All three scFv-Fc antibodies had nanomolar range affinities against the HER2-positive JIMT-1 cells. However, the dissociation constants were higher than the K_D_ for trastuzumab indicating the scFv-Fc antibodies had slightly reduced affinity. Of the three engineered antibodies, the DM scFv-Fc had the best affinity but had ~2.5-fold and 4.2-fold weaker K_D_ and EC_50_ values than trastuzumab. Interestingly, the WT scFv-Fc binding to HER2 resulted in the poorest K_D_ and EC_50_ values, but the Bmax value was comparable to trastuzumab.

### 3.6. MD Simulations Reveal Varying Degrees of Conformational Reorganization in scFv-Fc Constructs

In the energy-minimized models, all three scFv-Fc antibodies adopt the classical ‘Y’ conformation characteristic of human IgG_1_, with fully extended hinges projecting the scFv units away from the Fc (Figure 5). The Fc is oriented downwards with a slight tilt. Both V_H_ domains of the scFv subunits are positioned inward towards the central axis of the scFv-Fc, while the V_L_ domains extend outward. As a result, all the scFv-Fc antibodies have 67° of angular separation between the scFv1 and scFv2 subunits. In addition, the separation angles between the scFv1 and Fc1 and scFv2 and Fc2 subunits are 105° and 151°, respectively (Figure 5). However, after 220 ns of MD simulations, significant structural reorganization occurs, with each scFv-Fc format undergoing distinct conformational changes (Figure 5).

The WT scFv-Fc exhibited substantial structural reorganization, characterized by pronounced displacement of scFv2 toward scFv1 (Figure 5b). This movement was initiated early in the simulation (Appendix A), leading to scFv2 crossing over and interacting with scFv1. Subsequently, scFv1 extended before contracting towards the scFv-Fc central pivot where it formed new interactions with both scFv2 and the Fc. These conformational changes resulted in an overall compaction of the scFv-Fc structure (Figure 5b). Specifically, scFv1 and scFv2 contracted toward the central pivot by 46.7% and 32.3%, respectively, relative to their starting configurations. Additionally, scFv1 underwent a notable shift of 37Å in the z-axis.

The primary driver of this compact configuration is the formation of new interdomain interactions. The cumulative interaction energy between the V_H_ and V_L_ domains increased by 148% for scFv1 and 198% for scFv2 compared to their initial conformations. The total interchain interaction energy increased by 172%. Notably, multiple bonds stabilized this compact structure, including strong hydrogen bonds between residues S85-E89 (V_H_ domain of scFv1) and S149-D152 (V_L_ domain of scFv2). The most significant interaction was a salt bridge and hydrogen bond between E89 (scFv1) and K238 (scFv2) (Figure 6a). In comparison, the intra-domain interactions were very close for both antibody chains between the start and MD clustered conformations. These results indicate that the WT scFv-Fc formed a compact conformation through a notable scFv reorganization and a hinge contraction resulting in significant interchain binding and angular changes.

The spatial reorganization of the SM scFv-Fc deviated significantly from that of the WT scFv-Fc. The most significant structural change was whereby both scFvs were positioned with their CDRs facing approximately 180° in opposite directions (Figure 5c). Additionally, scFv1 and scFv2 contracted by 23.2% and 41.3%, respectively, compared to their starting positions. The notable structural feature of the SM scFv-Fc was the strong stabilization of the scFv linkers, which formed eight hydrogen bonds connecting the V_H_-V_L_ domains of scFv1 and scFv2 (Figure 6b). The two linkers interacting and the subsequent stabilization occurred very early in the simulation (Appendix A) and indicated this movement prevented further SM scFv-Fc reorganization and maintained the conformation within the same plane as the pivot point (Figure 5c). Consequently, the interactions between the V_H_ and V_L_ domains were considerably reduced, decreasing by 60% and 54% for scFv1 and scFv2, respectively, and suggest a destabilization of the overall scFv pairing. Additionally, the angles between the scFv and Fc for each chain were slightly widened. These results suggest that while the SM scFv-Fc adopted a more upright conformation compared to the contracted WT scFv-Fc, the strong linker interactions contributed to increased scFv rigidity.

Among the three scFv-Fc antibodies, the DM scFv-Fc exhibited the least structural reorganization at the end of its MD simulation (Figure 5d). Both scFv1 and scFv2 contracted towards the pivot point at ~50 ns in the simulation (Appendix A) similar to the WT and SM scFv-Fc antibodies. However, both scFvs were able to extend, with scFv2 notably further elongated than scFv1, where it also pitched forward in the z-axis (Figure 5d). This reconfiguration occurred at ~175 ns, where the DM scFv-Fc reached a second stabilization phase. At the end of the MD simulation, it appeared as if the scFvs were again contracting towards the center. Importantly, while interactions between scFv1 and scFv2 were present, based on the MD clustered structure, they were significantly weaker compared to those observed in the WT and SM scFv-Fc antibodies (Figure 6a,b). Additionally, unlike the other scFv-Fc variants, the interchain interactions in DM scFv-Fc decreased by ~23%, rather than increasing. These results indicate that the DM scFv-Fc maintained an overall lower degree of interchain interactions, suggesting a superior balance between stability and flexibility compared to the other variants.

Despite the substantial conformational rearrangements observed in the scFv-Fc antibodies, a detailed analysis of the CDRs revealed surprising spatial conservation. Using the crystal structure of the trastuzumab Fab in its non-HER2-bound state (PDB: 6BHZ) as a reference [30], the scFvs from the MD-clustered scFv-Fc antibodies were superimposed onto the V_L_ and V_H_ domains of 6BHZ, resulting in an average RMSD of <2 Å. This finding was further supported by domain-level RMSD plots, which showed that most structural motion originated from the V_L_ domain of the scFv2 and Fc chain 2 (Appendix A). This pattern was consistent with the increased elongation mobility of scFv2 for all engineered scFv-Fcs, albeit at different recurrent rates (Appendix A). In contrast, the V_H_ and V_L_ domains of scFv1 exhibited very little deviation (1–1.5 Å). The residues from Fc chain 1 exhibited intermediate deviations, most likely due to their dimerization with Fc chain 2. Overall, deviations within individual domains were low, with a maximum of ~3.5 Å. Taken together, these results suggest that despite large-scale structural reorganization at the whole-antibody level, the antigen-binding sites remained stable, and intra-domain unfolding was minimal.

### 3.7. MD Simulations and Impact on scFv-Fc Structural Dynamics

We utilized three standard statistical metrics, RMSD, RMSF, and Rg, to assess the structural dynamics and compactness of the engineered scFv-Fc antibodies over the simulation time. The RMSD plots representing the ɑ-carbon atom deviations between the original energy minimized and the clustered MD conformations revealed distinct patterns among the variants. All three scFv-Fc antibodies exhibited stabilizing trajectories after 50 ns with deviations reaching ~20–25 Å from their original energy-minimized positions, with the WT and DM antibodies experiencing slightly increased deviations at the end of the simulation (Figure 7a). The SM scFv-Fc showed an overall lower deviation profile, exhibiting consistent RMSDs at ~20 Å, which is indicative of a relatively more rigid conformation.

As previously described, a domain-level RMSD analysis strongly indicated that intra-domain fluctuations were marginal (Appendix A). The MD simulation movies revealed that the scFv-Fcs most likely underwent large molecular weight oscillating structural reconfigurations. For example, it was observed that each scFv-Fc eventually returned to a configuration closely resembling configurations very early in the simulations (Figure 7b). The final structures at 220 ns, when superposed onto their counterpart structures at these early time points, exhibited RMSD values of ~9–14 Å, which is within the cut-off range of 10 Å for evaluating the quality of superposing two identical but predicted structures for large molecular weight proteins [38].

Interestingly, when the RMSD plots were calculated based on removing the first 20 ns, the DM scFv-Fc displayed a biphasic deviation pattern, where RMSD remained ~15 Å from 60 to 150 ns, increased to >20 Å, and then sharply decreased towards the end of the 220 ns simulation (Appendix A). Additionally, the SM scFv-Fc RMSD profile indicates a very stable structure, while the WT scFv-Fc has a distinct steady, but highly deviated state.

The RMSF values, which measure residue-level flexibility, provided additional insight into the differential mobility of the scFv-Fc domains (Figure 8a). In general, the residue ɑ-carbon atoms of the V_H_ and V_L_ domains of scFv2 fluctuated more than the domains of scFv1 for all the scFv-Fc antibodies. The scFv2 residues for the WT scFv-Fc exhibited the greatest fluctuations reaching ~20 Å. In contrast, the SM scFv-Fc displayed uniformly low RMSF values across all domains. The DM scFv-Fc exhibited intermediate fluctuations, where scFv2 fluctuated substantially more than scFv1.

The Rg serves as a global indicator of protein compactness and aids in determining whether the scFv-Fc remained folded despite significant domain motion. The Rg values were consistent at ~75 ns for the WT and SM scFv-Fcs (Figure 8b). The WT scFv-Fc showed the most compactness, aligning with its MD-clustered configuration (Figure 5b). The SM scFv-Fc exhibited the least compactness, which aligned with its elongated MD-clustered structure (Figure 5c). Notably, the DM scFv-Fc displayed an oscillating gyration pattern, where Rg values increased from ~35–40 Å to ~45 Å and back down twice within the 220 ns simulation. This matched very closely to the biphasic RMSD profile in the Appendix A, Figure 6, and the early and late superposed configurations in Figure 7b.

The RMSD, RMSF, and Rg profiles ultimately support the clustered MD conformations (Figure 5) and the binding affinity results (Table 2). The high RMSD values (~20–25 Å) for the WT scFv-Fc aligned with its substantial structural deformation, while the high RMSF of scFv2 relative to scFv1 suggests that scFv1 was immobilized, contributing to its contracted conformation (Figure 5b). The Rg profile further supports that the WT scFv-Fc is highly compact. Despite its compactness, the MD simulation revealed transient scFv1 extensions (Appendix A), suggesting that both binding sites are intermittently available to bind HER2. This structural flexibility likely explains why the WT scFv-Fc maintained a Bmax very close to trastuzumab.

The SM scFv-Fc had the lowest RMSD and RMSF values yet the highest Rg deviations, which is consistent with its rigid and extended conformation (Figure 7 and Figure 8). Its configuration is due to strong linker–linker hydrogen bonding, locking both scFvs in opposite-facing directions (Figure 6b). This structural configuration with limited conformational flexibility correlates with its poor binding affinity and Bmax.

In contrast, the distinct profiles displayed by the DM scFv-Fc highlight a unique dynamic conformational adaptability highlighted by its oscillating RMSD (Appendix A) and Rg (Figure 8b) profiles. The DM scFv-Fc also had the least interchain interactions. Based on its excellent binding performance, these results suggest it could maintain a balance between conformational stability and flexibility, allowing for more effective HER2 binding compared to the WT and SM scFv-Fcs.

## 4. Discussion

This study investigated the biochemical and structural behaviours of trastuzumab-derived engineered scFv-Fc constructs and aimed to determine the significance of HER2 binding performance and to infer their future potential as therapeutic agents. First, it was found that the V_H_-V_L_ orientation was less prone to aggregation compared to the V_L_-V_H_ orientation. Second, high expression levels were challenging to achieve but scalable production was feasible. Third, the fragments were observed to be highly sensitive to purification and dimer purities varied. Fourth, thermal stability experiments revealed that all fragments exhibited similar melting points, albeit lower than trastuzumab. Fifth, flow cytometric assays on JIMT-1 cells demonstrated that the DM scFv-Fc antibody displayed the overall best affinity for HER2, albeit clearly weaker than trastuzumab. Sixth, MD simulations revealed that the WT and SM scFv-Fc antibodies underwent severe stochastic reorganizations compared to the DM scFv-Fc antibody. Seventh, mutations in the Fc domain showed no influence on structural configuration and stability. Finally, contact energy comparisons between energy-minimized and clustered MD scFv structures indicated that inter-antibody chain interactions drove large-scale reorientations, favouring scFv-Fc compactization. Taken together, the Fc does not appear to provide improved stability to an engineered antibody of a traditional scFv-Fc format, as was typically thought in the field. Additionally, although intra-domain unfolding was marginal and the scFv binding site CDRs remained stable, the scFv-Fc format is subject to significant large-scale structural reorganization, causing reoriented rigid conformations of the scFv subunits that limit bivalent binding.

Antibody engineering approaches have evolved tremendously during the past decades. The methods of computational protein design for engineering highly stable scFvs have been approached in earnest [39,40]. While computer modelling helps guide design and allows for identifying and analyzing potential weak points, experimental testing remains necessary to establish the clinical potential of engineered antibodies. The scFv format is popular due to its simple engineering design concept and its ability for fast accumulation and deep penetration into target tumours. However, its small size results in extremely short half-lives in plasma, which is a drawback if absolute high tumour uptake is required to evoke effective anti-tumour responses [41]. More importantly, for therapeutic design, we must consider that scFvs decrease thermal stability and potential to aggregate [42] in the context of larger-sized formats, which is an area that has not received sufficient attention.

It is currently accepted that the genetic fusion to Fc domains of IgG_1_ antibodies offers a reliable strategy to improve the overall stability of scFv-based therapeutics by capitalizing on dimerization due to interchain interactions between residues in the opposing hinge and C_H_3 domains. Despite their increased size, which counters the rapid tumour accumulation rate of scFvs, these fusion fragments still maintain faster tissue penetration and blood clearance compared to classical larger mAbs. Additionally, their half-life can be finely tuned through strategic point mutations within the Fc domain, particularly at amino acids I253, H310, H433, and H435 [43]. These residues serve as critical binding sites located at the interface of the C_H_2 and C_H_3 domains, where interaction with the FcRn receptor occurs. Introducing mutations at these key positions enables precise modulation of the disruption level in the interaction between the Fc domain and FcRn, leading to a diverse array of pharmacokinetic and tumour uptake profiles [44,45,46]. Therefore, this approach has been mostly viewed as promising to potentially enable these types of agents to reduce exposure times to healthy tissue while enhancing tumour-targeting specificity and therapeutic efficacy.

It is well known that the high degree of IgG conformational flexibility is vital for effective antigen binding by the CDRs. What is less appreciated but also equally important for effective antigen binding is that the V_H_ and V_L_ domains undergo conformational changes independently of each other as they form weak interactions for their dimerization. This independence allows for flexibility in the arrangement of these domains and the CDRs, which is important for antigen recognition and binding by an antibody. ‘Rewiring’ the connectivity of the V_H_ and V_L_ domains via an scFv format has been shown to change their dynamics and geometric orientation of the antigen binding site [33,47] and affect binding strength [32,33,47]. Engineered antibody fragments employing the V_L_-V_H_ orientation have been shown to have increased flexibility but reduced antigen specificity compared to the parental antibodies [36]. Because the CDR3 loop of the V_H_ domain is essential for antigen binding, it was previously reported that in a V_H_-V_L_ orientation, the flexible linker will not interfere with the CDR3 loops and, hence, limit the possibility of adversely interfering with antigen binding [48]. In this study, we observed that the linker was the main driver for the upright and rigid conformation for the SM scFv-Fc (Figure 6b). The two linkers from each scFv subunit formed multiple hydrogen bonds, which caused the binding domains to face approximately 180° in opposite directions (Figure 5c).

MD simulation provided key insights into how structural conformations influenced the binding performance of the scFv-Fc antibodies. The WT and SM scFv-Fcs exhibited the greatest structural reorganizations and correspondingly had the poorest binding affinities against HER2+ JIMT-1 cells (Table 2). In the WT scFv-Fc, scFv1 became stabilized between scFv2 and the Fc (Figure 5b), which most likely restricted its accessibility to HER2. Similarly, in the SM scFv-Fc, both scFv subunits were stabilized in a configuration that also likely reduced bivalent HER2 binding. In contrast, the DM scFv-Fc had both of its scFv subunits elongated, which supported maintaining the bivalent binding of HER2. Nevertheless, the WT scFv-Fc displayed a Bmax similar to trastuzumab, suggesting that despite its compact structure, it could still recognize HER2 with the same number of available binding sites. This raises the possibility that even the contracted and stabilized scFv subunit in the WT scFv-Fc might retain some degree of conformational flexibility, enabling it to extend upon antigen engagement and achieve bivalent binding.

In contrast, the SM scFv-Fc exhibited a higher Bmax, which may indicate that the strong interactions between the scFv linkers lock the two scFvs in a rigid conformation, preventing effective bivalent binding. This rigid structure likely requires a greater number of SM scFv-Fc molecules to saturate all HER2 binding sites. The DM scFv-Fc also displayed a high Bmax, which is more difficult to explain given its relatively stable structure. One possible interpretation is that while the scFvs remain mostly extended, one scFv underwent transient contraction, which could have reduced its availability for HER2 binding at certain time points. Indeed, MD simulations showed that the DM scFv-Fc momentarily contracted before re-extending (Appendix A). Taken together, these findings emphasize the importance of structural flexibility in scFv-Fc design and highlight the need to balance stability with dynamic adaptability to ensure efficient antigen binding in antibody fragment-based therapeutics.

The DSF results suggest that the introduced mutations in the Fc region, represented by the SM and DM variants, did not substantially alter the thermal stability of the scFv-Fc antibodies compared to the WT scFv-Fc counterpart. The overall weaker stability compared to that previously reported for trastuzumab [37] indicates the scFv-Fc format itself is less stable. Further investigations into other aspects of stability, such as long-term storage stability or resistance to denaturation under different conditions, would provide a more comprehensive understanding of the impact of these mutations on the overall stability of the scFv-Fc antibodies. Nonetheless, 5-week storage of these agents at 4 °C revealed that the dimer population remained stable (Figure 2b).

The combined in silico and biochemical studies provided essential insights into scFv-Fc development, revealing this ‘traditional’ engineered antibody fragment format does not fully maintain the classical Y-shape. Instead, MD simulations demonstrated that scFv subunits undergo stochastic movement, leading to significant fluctuations in their spatial positions. This dynamic behaviour increased interchain interactions that restricted motion and most likely limited HER2 accessibility, supported by the weaker binding affinities. Notably, the spatial positioning of the CDRs remained largely unchanged and further supports that large-scale structural rearrangements and not intra-domain unfolding were the root causes for poor binding performances. While high RMSF values were observed, these did not reflect structural instability. Rather, they signify tolerable flexibilities, which can possibly positively (DM) or negatively (WT and SM) influence HER2 binding. Overall, this study demonstrated that the scFv-Fc format introduces inherent dynamic motions that negatively impact overall binding efficiency, with the linker–hinge segment connecting the scFv to the Fc as the primary source of instability. Although the overall conformations of the scFv-Fc are stochastic, the DM scFv-Fc exhibited the most favourable properties for preclinical therapeutic applications.

Future development efforts will likely require advanced computational protein design strategies, including artificial intelligence, to enhance scFv-Fc stability and mitigate stochastic conformational fluctuations. The use of computational modelling in this study proved essential for precisely identifying structural regions for optimization and aligned well with the experimental biochemical results. Lastly, this work underscores the importance of an interdisciplinary approach to antibody engineering, integrating computational modelling with experimental validation to develop next-generation antibody fragments for therapeutic applications.

## 5. Conclusions

By combining computational modelling with biochemical analysis, this study revealed that trastuzumab-based engineered scFv-Fc antibody fragments exhibit distinct and stochastic global dynamics driven by structural design rather than Fc mutations. Although all constructs retained nanomolar HER2 affinity, their expression profiles, dimer stability, and domain orientations varied significantly, with the scFv-Fc architecture itself, particularly the scFv–hinge, emerging as key determinants of performance. The WT and SM scFv-Fcs adopted either overly compact or elongated and rigid conformations that limited effective bivalent binding, while the DM scFv-Fc maintained a more favourable balance of dynamic flexibility and compactness, which enabled it to exhibit superior HER2 binding. These results clarify how stochastic structural reorganization, rather than fixed conformational states, shapes the functional behaviour of scFv-Fc fragments and provides a template for evaluating future similar engineered antibodies.

## Figures and Tables

**Figure 1 biomolecules-15-00606-f001:**
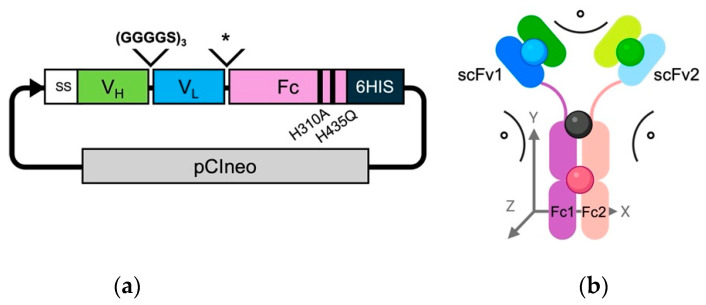
Schematic representation of the engineered scFv-Fcs. (**a**) Schematic of pCIneo plasmid containing the construct encoding the scFv-Fc format. SS = IL-2 secreting signal and 6HIS = 6 histidine tag; * represents the human antibody hinge VEPKSCDKTYTCPPCP; and black bands represent H310A and H435Q mutations. (**b**) Schematic of the scFv-Fc antibody in an energy-minimized state. The antibody chains are colour-coded with darker shades of green and blue representing the V_H_ and V_L_ domains for scFv chain 1, respectively. The lighter shades of green and blue represent the V_H_ and V_L_ domains for scFv chain 2, respectively. The Fc1 and Fc2 domains are coloured purple and peach, respectively. The center pivot for calculating angles was the equidistant point (black sphere) of the P238 residues from Fc1 and Fc2. The center points for scFv1 (blue sphere), scFv2 (green sphere), and the Fc dimer (red sphere) were determined by taking the mean for all residue 3D spatial coordinates. Angles were calculated for the relative positions in the XY-plane between scFv1 and Fc1, scFv2 and Fc2, and scFv1 and scFv2 relative to the pivot point. Any distance changes in the z-direction are noted in Å.

**Figure 2 biomolecules-15-00606-f002:**
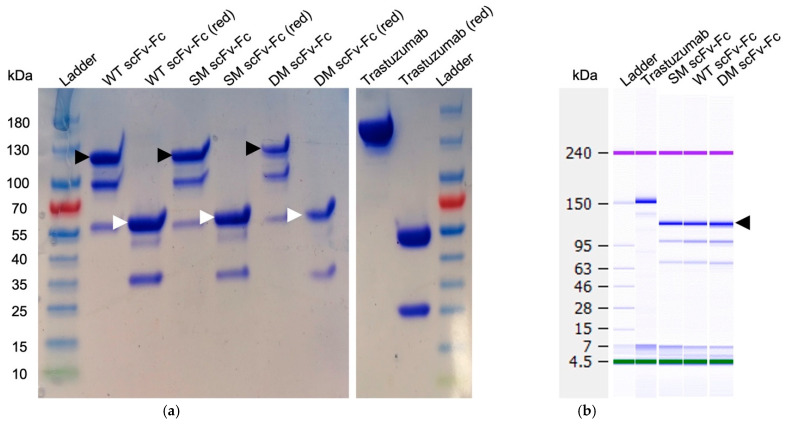
Gel mobilities of the engineered antibodies. (**a**) SDS-PAGE of engineered antibodies. The molecular weight for the scFv-Fc antibodies was ~120 kDa for dimers while monomers under reduced conditions were ~55 kDa. Black arrows represent dimers; white arrows indicate monomers after reduction (red) with DTT. Additional bands demonstrate the partial degradation of the scFv-Fcs. (**b**) Bioanalyzer results of the same engineered antibody fragments after 5-week storage at 4 °C under non-reducing conditions. Trastuzumab antibody was used as a loading control for both gels. SDS-PAGE original images can be found in Appendix A.

**Figure 3 biomolecules-15-00606-f003:**
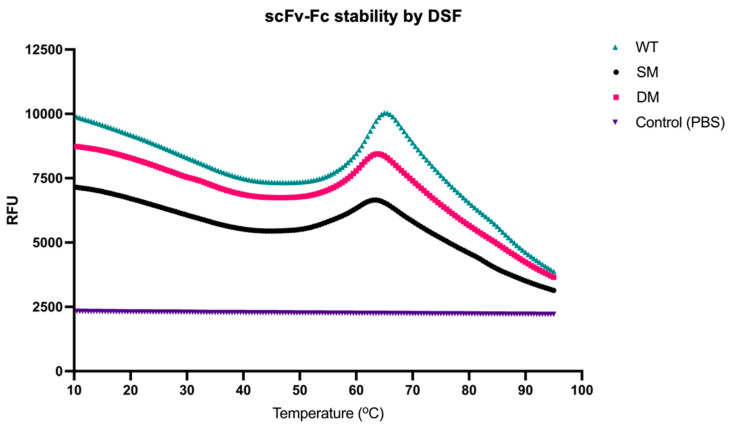
scFv-Fc stability by DSF. The melting temperatures (Tm) were evaluated by DSF from 10 °C to 95 °C.

**Figure 4 biomolecules-15-00606-f004:**
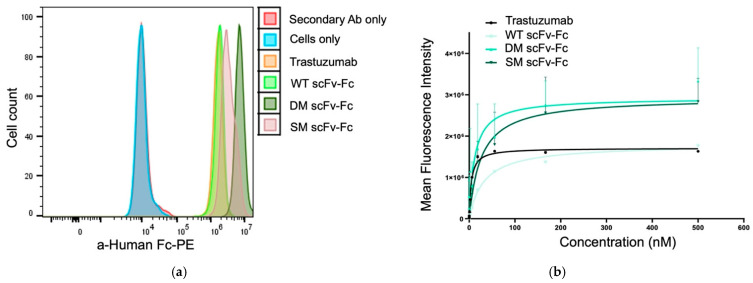
HER2 binding characteristics of the scFv-Fc antibodies. Binding to the HER2 receptor was evaluated by indirect flow cytometry where JIMT-1 cells were incubated with trastuzumab and the scFv-Fc antibodies at (**a**) saturating or at (**b**) increasing concentrations. After washing, the cells were stained using a secondary anti-human Fc antibody.

**Figure 5 biomolecules-15-00606-f005:**
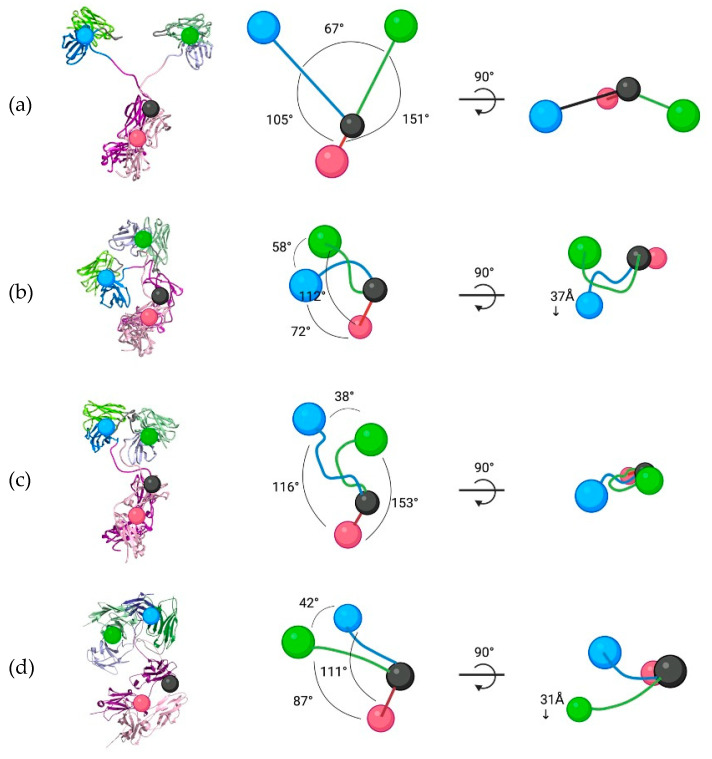
Structural organizational changes between the energy-minimized and clustered MD structures. The spatial and structural configurations of (**a**) starting scFv-Fc structures and the clustered MD conformations for the (**b**) WT, (**c**) SM, and (**d**) DM scFv-Fc antibodies. The conformations are based on the center points for the scFv1 (blue), scFv2 (green), and Fc dimer (red) as seen on the computationally modelled structures.

**Figure 6 biomolecules-15-00606-f006:**
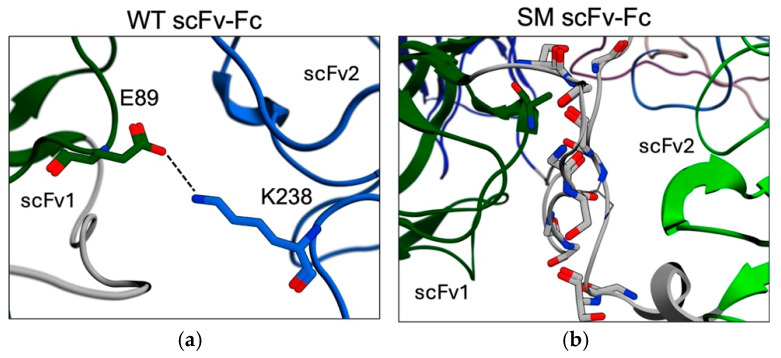
The interactions and structural changes for the clustered MD structures. (**a**) Interactions between the V_H_ and V_L_ domains of scFv1 and scFv2, respectively, and the strongest interaction was through the E89 and K238 hydrogen bond and salt bridge (dashed line). (**b**) Multiple contacts between the two linkers (grey) for the scFv subunits.

**Figure 7 biomolecules-15-00606-f007:**
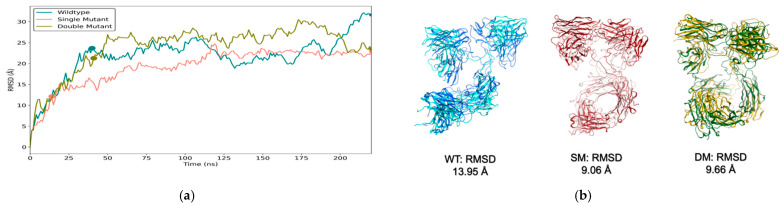
RMSD plots from MD simulations. The scFv-Fc antibodies were subjected to 220 ns of MD simulations and the (**a**) RMSD (Å) was plotted over time. (**b**) The corresponding structures from the final frame at 220 ns for the WT (blue), SM (crimson), and DM (green) scFv-Fcs superimposed against their structurally closest frames during the simulation. The closest frames for the WT, SM, and DM scFv-Fcs were at 40.26 ns (cyan), 14.08 ns (salmon), and 40.70 ns (olive), respectively. The RMSDs between the corresponding structures for each scFv-Fc are shown.

**Figure 8 biomolecules-15-00606-f008:**
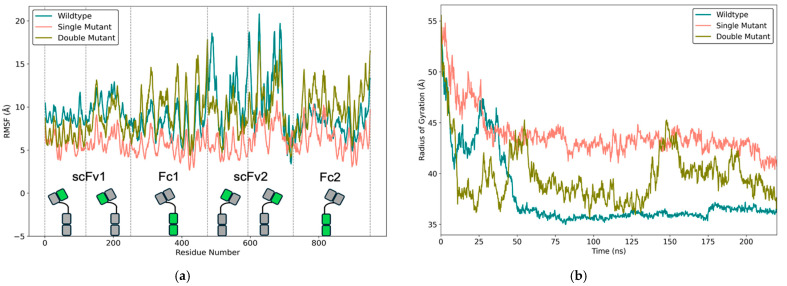
RMSF and Rg plots from MD simulations. (**a**) RMSF of α-carbon atoms for each scFv-Fc, plotted against residue number. Vertical dashed lines indicate domain boundaries, and the specific region related to the amino acid number for the scFv-Fc is highlighted in green. (**b**) Rg over the same simulation time window.

**Table 1 biomolecules-15-00606-t001:** Antibody fragment yields after being expressed in different cell lines.

Cell Line	Yield (µg/mL)	Days of Culture
CHOK1	0.088–1.060	7
HEK293T	0.063–0.543	7
ExpiCHO-S	5.866–19.066	8
293F	700–850	5

**Table 2 biomolecules-15-00606-t002:** Dissociation constant (K_D_) (affinity to HER2), binding potential (Bmax), and half-maximal effective concentration (EC_50_) for trastuzumab and the tested engineered antibodies.

Antibody Name	K_D_ (nM)	Bmax	EC_50_ (nM)
Trastuzumab	4.5 ± 0.1	1,708,486 ± 10,135	4.1 ± 0.1
WT scFv-Fc	32.5 ± 1.0	1,795,860 ± 14,608	31.9 ± 0.7
SM scFv-Fc	26.0 ± 6.8	2,929,382 ± 169,638	24.3 ± 4.1
DM scFv-Fc	12.0 ± 8.9	2,921,012 ± 307,450	17.2 ± 7.5

## Data Availability

The original contributions presented in this study are included in the article/Appendix A. Further inquiries can be directed to the corresponding author(s).

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
