# Peer review of "Integrating Biochemical and Computational Approaches Reveal Structural Insights in Trastuzumab scFv-Fc Antibody Engineering"

_biomolecules, 2025, doi:10.3390/biom15050606_

Round 1
Reviewer 1 Report
Comments and Suggestions for Authors
The manuscript by Bednova et al. presents a comprehensive study on trastuzumab-derived scFv-Fc antibodies, assessing the impact of single and double mutations on various biopharmaceutical parameters through both experimental and computational approaches. Specifically, the authors used thermal stability (Tm) assays, aggregation assessments, binding kinetics and molecular dynamics (MD) simulations to gain structural insights. The paper is clearly structured and addresses key aspects effectively. However, there are few computational aspects that require clarification and additional analysis before publication:
- It is unclear to me why 1HZH was used as a template in MOE when there are several Trastuzumab antibody structures available in the PDB database.
- The citation for the 1HZH appears to be incorrect. According to the PDB database, the correct PMID is 11498595.
- The RMSD analysis presented in the manuscript indicates that the wild-type antibody reaches equilibrium at approximately 80 ns, whereas the single and double mutants do not appear to achieve equilibrium even after 220 ns of simulation time. There is significant deviation in RMSD especially after 200 ns. Authors should consider extending the MD simulations for these mutants to justify their observations.
- High RMSF was primarily observed in the VL and VH regions. Authors should compare the initial modeled and final MD-simulated ScFv structures to experimental wild type structure to show the structural differences.
- The Fab region of antibodies also contains disordered regions. It would be valuable if the authors provided explicit commentary or analysis on these disordered regions and correlated these findings with their own RMSF data (see PDB id 1n8z for example).
Author Response
Dear Reviewers,
Thank you for your valuable feedback on our manuscript. We have taken your comments into careful consideration and made significant revisions to the manuscript based on your comments and suggestions. As a result of these revisions, we believe we have significantly improved the robustness of the study and increased the clarity of the data presented and the discussion of the findings.
We sincerely appreciate your constructive input, which has contributed to strengthening the quality of the work.
Reviewer 1
Comments and Suggestions for Authors
The manuscript by Bednova et al. presents a comprehensive study on trastuzumab-derived scFv-Fc antibodies, assessing the impact of single and double mutations on various biopharmaceutical parameters through both experimental and computational approaches. Specifically, the authors used thermal stability (Tm) assays, aggregation assessments, binding kinetics and molecular dynamics (MD) simulations to gain structural insights. The paper is clearly structured and addresses key aspects effectively. However, there are few computational aspects that require clarification and additional analysis before publication:
Response.
Thank you for the kind feedback. We believe that this work enhances the current understanding of the relationship between scFv-Fc engineering and stability, expression, and binding performance. The revised manuscript includes substantially more computational work, which we aimed at addressing your comments in this specific area.
General comments
- It is unclear to me why 1HZH was used as a template in MOE when there are several Trastuzumab antibody structures available in the PDB database.
Response.
Thank you for your detailed attention as we see the original description is confusing. MOE was used to build in silicoversions of the trastuzumab scFv-Fcs. This was performed by selecting PDB: 1HZH as it is one of the rare structures of a fully intact human IgG1 antibody. Our goal was to use the hinge-Fc as its sequence was identical to the sequence we used for the hinge-Fc of the WT scFv-Fc. As the 1HZH antibody was specific for HIV-1, we deleted the Fab domains. MOE contains a state-of-the-art protocol for constructing human scFvs. It is based on homology modeling and incorporates several trastuzumab Fab fragments. Once the scFv models were generated, we built the additional short amino acid extension that connects the scFv to the hinge of the Fc. The structure was then saved and two additional versions duplicated containing the mutations for the SM and DM scFv-Fcs. All three in silico scFv-Fc models were then energy minimized and ready for MD simulations. We have clarified this in the revised text in the Methods, section 2.7.
- The citation for the 1HZH appears to be incorrect. According to the PDB database, the correct PMID is 11498595.
Response.
Thank you for catching this error. Indeed, the incorrect Saphire et al., reference was used. The correct Saphire et al., Science, 2001, reference [25] is now in place.
- The RMSD analysis presented in the manuscript indicates that the wild-type antibody reaches equilibrium at approximately 80 ns, whereas the single and double mutants do not appear to achieve equilibrium even after 220 ns of simulation time. There is significant deviation in RMSD especially after 200 ns. Authors should consider extending the MD simulations for these mutants to justify their observations.
Response.
We appreciate your thoughtful observation regarding the RMSD behavior of the single and double mutant (SM and DM) scFv-Fc constructs. While it is correct that the WT scFv-Fc appeared to reach a steady-state RMSD around 80 ns, we respectfully clarify that although the SM and DM scFv-Fcs did not display a single convergent equilibrium, they exhibited variant-specific equilibria.
In our revised manuscript (Section 3.7), we now include a more detailed analysis. First, we calculated the RMSD profiles based off the 0 ns start point (Fig. 7A). Previously, we showed the RMSD plots based on a 20 ns start point, which we rationalized would be better to eliminate the first 20 ns (now Suppl. Fig. 6). By starting at 0 ns, all the scFv-Fcs converge to approximately the same equilibria (Fig. 7A).
However, our deeper analysis revealed that indeed the scFv-Fcs, more so DM, undergo recurrent return-to-statebehaviors, supported by superposition analysis (Fig. 7B), where the final structures at 220 ns closely align (RMSD ~9–14 â„«) with early simulation conformations. This behavior is further corroborated by biphasic or oscillating RMSD and Rg profiles (Suppl. Fig. 6 and Fig. 8B), especially in the DM variant. These dynamics suggest the systems are sampling from a stable conformational ensemble rather than remaining in transition.
We also performed a domain-level RMSD analysis, which showed that the domains remained low (≤3.5 â„«), indicating no signs of intra-domain unfolding or instability. The increased RMSD in the DM scFv-Fc near the simulation end reflects reversible global hinge reorganization rather than a drift toward unfolding or instability. Therefore, extending the simulation beyond 220 ns would likely reveal continued stochastic reorganization rather than yielding substantially different mechanistic insights.
We have revised the Results section to better emphasize these findings and to clarify that the system behavior represents dynamic equilibrium and not a failure to equilibrate. We believe this justifies the simulation length and supports the robustness of our conclusions.
- High RMSF was primarily observed in the VL and VH regions. Authors should compare the initial modeled and final MD-simulated ScFv structures to experimental wild type structure to show the structural differences.
Response.
We thank you for this helpful suggestion. As noted in Figure 8A, elevated RMSF values were observed in the VL and VH domains for scFv2. To assess whether these fluctuations translated into meaningful structural differences, we performed direct superpositions of the MD-clustered scFv1 and scFv2 structures onto the crystal structure of an unbound wild-type trastuzumab Fab (PDB:6BHZ). This procedure is described in the Methods (Section 2.8) and the results in Section 3.6. The results reveal that the RMSD between the MD-clustered scFvs and the experimental Fab VL and VH domains was <2 â„« across all scFv-Fcs, and indicates strong intra-domain structural conservation.
- The Fab region of antibodies also contains disordered regions. It would be valuable if the authors provided explicit commentary or analysis on these disordered regions and correlated these findings with their own RMSF data (see PDB id 1n8z for example).
Response.
We thank you for this insightful comment. While we acknowledge that certain regions within the Fab can exhibit intrinsic disorder (as reported in structures such as PDB 1N8Z), our RMSF analysis did not reveal evidence of localized disorder beyond expected loop flexibility. While we agree that this is an interesting topic, the impact of our study was primarily on global conformational dynamics and scFv-Fc domain organization, rather than detailed disorder mapping. Furthermore, our study revealed that reorganization differences were not due to intra-domain unfolding, which was very minimal. Given our data and the significance of the results, we chose not to further interrogate this aspect to maintain the study’s focus.
Reviewer 2 Report
Comments and Suggestions for Authors
The manuscript is well designed and neatly drafted. The manuscript is reviewed and the following minor concerns are to be addressed prior to publication.
1. The melting temperature values significantly drop upon mutants, which can be explored computationally to create necessary mutations in the engineered antibody to match the standard melting point of 80 °C.
2. The Control PBS is referred to in Figure 3; however, no plot or data points are visible.
3. RMSD in the range of 15-20 Å is considerably high for structural studies. The present RMSD analysis shows the WT and DM scFv-Fc over 20 Å. Do these higher deviations actually relate to biological conditions?
4. The RMSD plot shows greater deviations up to 60 ns, and henceforth, only WT has shown a stable pattern , whereas the SM and DM have recorded an increasing pattern. The mutants (SM and DM) endpoints of the simulations are not stable. Thus, extending the simulation up to 500 ns would give detailed insight into the structural movements and conformational behaviour.
5. The RMSF profile of the WT, SM, and DM structures shows complete fluctuations in the range of 10â„« (between 5 and 15â„«) on average, which represents that the protein structure is not stable. Will it have significant biological action with higher RMSF?
Author Response
Dear Reviewers,
Thank you for your valuable feedback on our manuscript. We have taken your comments into careful consideration and made significant revisions to the manuscript based on your comments and suggestions. As a result of these revisions, we believe we have significantly improved the robustness of the study and increased the clarity of the data presented and the discussion of the findings.
We sincerely appreciate your constructive input, which has contributed to strengthening the quality of the work.
Reviewer #2
The manuscript is well designed and neatly drafted. The manuscript is reviewed, and the following minor concerns are to be addressed prior to publication.
Response.
Thank you for your kind assessment of our manuscript and for the helpful suggestions. We have carefully addressed your concerns and revised the manuscript accordingly. We appreciate your time and effort in reviewing our work.
General comments
- The melting temperature values significantly drop upon mutants, which can be explored computationally to create necessary mutations in the engineered antibody to match the standard melting point of 80 °C.
Response.
We are grateful for this insight as introducing stabilizing mutations could improve the overall properties of these scFv-Fc fragments. Although this beyond the scope of this current work where we focus on the traditional scFv-Fc format, we will incorporate this approach into future research.
- The Control PBS is referred to in Figure 3; however, no plot or data points are visible.
Response.
Thank your catching this error in Figure 3. We have provided updated Figure 3 in the main text.
- RMSD in the range of 15-20 Å is considerably high for structural studies. The present RMSD analysis shows the WT and DM scFv-Fc over 20 Å. Do these higher deviations actually relate to biological conditions?
Response.
Thank you for raising this important point. While RMSD values in the 15–20 Šrange may appear high in traditional structural studies, it is important to note that our analysis involves large multi-domain, flexible antibody fragments undergoing domain-level reorganization. As revised in the revised Section 3.7, the high RMSD primarily reflects global scFv-Fc rearrangements, not local unfolding or loss of structural integrity. This is supported by domain-level RMSDs remaining ≤3.5 Šand consistent CDR alignment with the crystal structure (PDB: 6BHZ). This is further supported, by the domain-level RMSD results, that show the domains themselves have only very small deviations (Suppl. Fig. 5).
These global reorientations are likely reflective of the inherent flexibility of antibody domains, particularly in engineered formats such as scFv-Fcs. Moreover, such motions may be biologically relevant, as they could influence epitope accessibility, bivalent binding potential, and Fc-mediated interactions as this study strongly suggests. We have clarified this important point in the text and emphasized that the observed RMSD values reflect flexible domain positioning within a folded, functionally competent scFv-Fc.
- The RMSD plot shows greater deviations up to 60 ns, and henceforth, only WT has shown a stable pattern, whereas the SM and DM have recorded an increasing pattern. The mutants (SM and DM) endpoints of the simulations are not stable. Thus, extending the simulation up to 500 ns would give detailed insight into the structural movements and conformational behaviour.
Response.
We appreciate your suggestion and agree that longer MD simulations can provide additional insights into conformational dynamics. Our primary goal was to identify meaningful structural rearrangements in the context of scFv-Fc design and the relationship with biochemical parameters. Originally, we showed that the scFv-Fcs did not converge to a single equilibrium. However, this RMSD analysis was performed based on the 20 ns start time, which was used to eliminate signal noise.
In our revised analysis (Section 3.7), we display the RMSD plot analyzed from the 0 ns start time, which shows that the three scFv-Fc do converge to approximately the same equilibria (Fig. 7A). We further noted that while the SM and DM scFv-Fcs exhibited increasing RMSD late in the trajectory, this behavior corresponded to reversible domain rearrangements, not a continuing divergence. This is supported by the superposition of early and late structures (Fig. 7B), the biphasic/oscillating RMSD profiles for from the original 20-220 ns RMSD (Suppl. Fig. 6) and Rg profiles (Fig. 8B), particularly for DM scFv-Fc that showed a “return-to-state” behavior. These structural behaviors all match well with the binding studies. Therefore, the 220 ns simulation sufficiently captures the critical essence of the structural behavior of these trastuzumab-based scFv-Fc designs. An extension to 500 ns would likely yield redundant transitions without substantially altering our conclusions.
- The RMSF profile of the WT, SM, and DM structures shows complete fluctuations in the range of 10â„« (between 5 and 15â„«) on average, which represents that the protein structure is not stable. Will it have significant biological action with higher RMSF?
Response.
Thank you for raising this important point. While RMSF values in the 5–15 Šrange may appear high, particularly in the scFv2 regions, these values reflect expected flexibility in solvent-exposed loops and hinge regions, not loss of structural integrity. This is supported by domain-level RMSDs ≤3.5 Šand structural superpositions of the CDRs with the trastuzumab Fab crystal structure (PDB: 6BHZ), which show minimal deviation (<2 Å), indicating no unfolding or distortion of the antigen-binding sites.
Functionally, this mobility may be biologically relevant rather than detrimental. The observed flexibility, especially in the DM scFv-Fc, likely facilitates favorable spatial orientations for HER2 binding, correlating with its superior affinity and Bmax values. In contrast, the SM variant, despite lower RMSF, displayed poorer binding, highlighting that rigid structures are not inherently more functional.
Together with the RMSD discussion for the above comment, these findings show that moderate flexibility, particularly when localized to variable domains, does not inhibit biological action but may enhance it by enabling adaptable binding orientations. We now emphasize this point more clearly in the revised Results and Discussion sections.
Round 2
Reviewer 1 Report
Comments and Suggestions for Authors
The authors have effectively addressed the majority of my concerns, and I believe the paper is suitable for publication in Biomolecules in its current form.